# Receiver Location Optimization for Heterogeneous S-Band Marine Transmitters in Passive Multistatic Radar Networks via NSGA-II

**DOI:** 10.3390/s25185861

**Published:** 2025-09-19

**Authors:** Xinpeng Li, Pengfei He, Jie Song, Zhongxun Wang

**Affiliations:** 1School of Physics and Electronic Information, Yantai University, Yantai 264005, China; xinpengli6@gmail.com (X.L.); hpf_972@ytu.edu.cn (P.H.); 2Shandong Provincial Laboratory of Data Open Innovation and Application for Advanced Smart Grid Technologies, Yantai University, Yantai 264005, China; 3Research Institute of Information Fusion, Naval Aviation University, Yantai 264005, China; songjie@csif.org.cn

**Keywords:** non-cooperative passive multistatic radar, detection performance analysis, nondominated sorting genetic algorithm II (NSGA-II), maritime monitoring

## Abstract

Comprehensive maritime domain awareness is crucial for navigation safety, traffic management, and security surveillance. In the context of an increasingly complex modern electromagnetic environment, the disadvantages of traditional active single-station radars, such as their high cost and susceptibility to interference, have started to surface. Due to their unique advantages, such as low cost, environmental sustainability (by reusing existing signals), and resilience in congested spectral environments, non-cooperative passive multistatic radar (PMR) systems have gained significant interest in maritime monitoring. This paper presents the research background of non-cooperative passive multistatic radar systems, performs a fundamental analysis of the detection performance of multistatic radar systems, and suggests an optimization method for the transceiver configuration of non-cooperative passive multistatic radar systems based on geometric coverage theory and a signal-to-noise ratio model. A multi-objective optimization model is developed, considering both detection coverage and positioning error, and is solved using the Non-dominated Sorting Genetic Algorithm II (NSGA-II). The optimization aims to find the optimal receiver location relative to a fixed configuration of four transmitters, representing common maritime traffic patterns. According to the simulation results, the multi-target genetic algorithm can be utilized to optimize the receiver position under the S-band radar settings used in this work. Compared to a random placement baseline, this can reduce the positioning error by about 8.9% and extend the detection range by about 15.8%. Furthermore, for the specific four-transmitter configuration and S-band radar parameters considered in this study, it is found that the best detection performance is more likely to be obtained when the receiver is placed within 15 km of the transmitters’ geometric center.

## 1. Introduction

Traditional active monostatic radars can face challenges in reliably detecting low-observability targets, such as non-cooperative vessels or those with small radar cross-sections (RCS), which is a common issue in maritime surveillance [1]. Because of their inherent benefits of low cost, strong survivability, and superior anti-jamming capabilities, non-cooperative passive multistatic radar (PMR) systems—which take advantage of illuminators of opportunity, such as digital broadcast transmitters, communication satellites, or commercial marine radars—have attracted a lot of attention in this context. The majority of commercial vessels now have Coastal Marine Radars (CMR) and Automatic Identification Systems (AIS) installed due to directives from the International Maritime Organization (IMO), which has made maritime surroundings especially rich in opportunistic signals [2]. The radar horizon is expanded by the raised installation of CMRs on large ships, such as tankers and cargo ships, and their narrow beamwidth and high transmit power make them ideal for integration into multistatic radar systems [3].

The special geometry between targets, passive receivers, and opportunistic transmitters (such shipborne radars) creates a non-cooperative PMR system. Without requiring ‘blanking’ Electronic Support Measures (ESM) receivers during transmission pulses, this architecture offers full passive detection capability with coverage comparable to monostatic systems, providing significant theoretical and practical value for contemporary maritime monitoring [4].

The core of non-cooperative passive detection lies in utilizing third-party electromagnetic signals to detect, locate, and track targets. In contrast to their cooperative counterparts, these systems must overcome the crucial technological obstacle of synchronizing location, time, and frequency without knowing the transmitted waveform beforehand. PMRs have several advantages over conventional active radars: (1) enhanced low-altitude coverage by mitigating multipath effects, (2) increased deployment flexibility and survivability due to receiver passivity, and (3) significantly lower network costs by utilizing existing radiators [5].

Previous research has validated the feasibility of this concept. Geometric configuration and ESM receiver sensitivity are critical elements for detection coverage, as demonstrated by Chong Sze Sing’s use of S-band signals from merchant ships for passive multistatic detection of maritime targets [6]. Likewise, a U.S. Naval Postgraduate School study used a frigate with EW/DF receivers in a PBR/PMR setup to investigate ideal geometries and quantify how system factors affect localization uncertainty [7]. Si et al. expanded on this by proposing a high-frequency hybrid technique for efficient ship target RCS simulation, which makes it easier to analyze the ideal system design [3].

However, conventional joint transmitter-receiver optimization techniques are made useless under non-cooperative restrictions, focusing only on optimizing receiver deployment. There is still a gap in the combined optimization of receiver location within a realistic, multi-transmitter geometric framework, even though previous research has mostly concentrated on optimizing transmitter placement or analyzing preset setups.

To address this gap, this paper makes a novel contribution by proposing a comprehensive optimization framework based on NSGA-II. In contrast to earlier research, our method simultaneously optimizes the receiver location in relation to a fixed four-transmitter configuration, which is a scenario that reflects common maritime traffic patterns. We create a multi-objective model that takes into account both detection coverage and localization accuracy using ship RCS data from [3]. In addition to showing notable performance improvements, our simulation results provide an important practical realization: the ideal receiver deployment is located inside a certain restricted region with respect to the geometric center of the transmitters.

The simulation scenario in this work, which utilizes emissions from commercial marine radars, is designed to explore the fundamental limits and optimization strategies of passive multistatic radar networks for civil maritime applications. These applications include maritime domain awareness, traffic management, search and rescue operations, and the monitoring of illegal activities within exclusive economic zones. It is crucial to emphasize that this research is conducted from a technical and methodological perspective. The practical implementation of such systems would, of course, be subject to strict international regulations, safety standards, and ethical guidelines governing the use of radio frequencies and the protection of civilian vessels. The ‘non-cooperative’ nature of the illuminators herein refers solely to the technical paradigm of not actively controlling the transmitters, and not to any intent to circumvent legal frameworks.

This is how the remainder of the paper is organized: The detection performance and localization concept of the PMR system are theoretically analyzed in Section 2, which also describes the multi-objective genetic algorithm and how it is used in the optimization model. The simulation procedure is explained and the outcomes are displayed in Section 3. Section 4 discusses the results, concludes the paper, and suggests directions for future research.

## 2. Materials and Methods

### 2.1. Analysis of Detection Performance of Non-Cooperative Multistatic Radar Systems

Numerous metrics, including target recognition ability, target placement accuracy, coverage range, and anti-interference ability, can be used to gauge a multistatic radar system’s detection capability.

Specifically, multistatic radar systems’ detection capability is primarily demonstrated by their capacity to find, identify, and continuously track targets within a specific orientation [8]. Detection range (also known as detection distance) is typically used to describe this performance. The following are typical characterization metrics:(1)The detection probability of the target by the detection system within a specific direction and distance range, with a focus on analyzing the influence of environmental factors and disturbances on the detection effect of the target—by comprehensively considering these factors, the target detection capability of radars in complex environments can be evaluated more accurately.(2)The maximum detection distance that the detection system can achieve under the given false alarm rate and detection probability—this method starts from the perspective of radar layout and configuration to analyze the overall performance of the system. By setting reasonable false alarm rates and detection probability thresholds, it assesses the maximum detection distance that the radar can achieve while maintaining high performance.

Although each of these two characterization metrics has a specific focus, taken as whole, they provide a thorough evaluation of multistatic radar detection capability. In real-world applications, suitable characterization metrics can be chosen according to particular needs and situations in order to assess and enhance radar performance.

#### 2.1.1. Multistatic Radar Detection Range

There are typically two ways to characterize a bistatic radar’s detection range: (1) Detection range under power limitation: By determining the minimum detectable signal power that the receiver can detect and substituting it into the radar equation, the maximum detection distance of the radar is calculated. (2) Range of detection under line-of-sight restrictions: The Earth’s surface limits the detection range of certain radar designs (such ground radars), which are primarily influenced by the line-of-sight distance and radar antenna height. The direct line-of-sight distance and antenna height typically restrict the detection range for targets that are beyond the horizon; with ground radar systems, this restriction is considerably more noticeable [9].

First, let us consider the detection range under power limitations. The minimum detectable radar signal from the radiation source is defined as follows:(1)$$S_{i\;\min}=kT_{R}NBF_{n}{\left(S/N\right)}_{o\;\min}$$
where the detecto ${\left(S/N\right)}_{o\;\min}$ is represented by the detection factor $D_{0}={\left(P_{R}/N_{0}\right)}_{o\;\min}$ and the receiving accumulation time is denoted by $t_{i}$, the distance product can be expressed as follows:(2)$${\left(R_{T}R_{R}\right)}_{\max}^{2}=\frac{P_{T}G_{T}G_{R}\lambda^{2}F_{T}^{2}F_{R}^{2}\sigma_{b}t_{i}}{{\left(4\pi \right)}^{3}kT_{R}BND_{0}L_{T}L_{R}L_{s}}$$

In the formula, $P_{T}$ represents the radar transmission power; $G_{T}$ is the gain of the transmitting antenna; $G_{R}$ is the gain of the receiving antenna; λ stands for radar wavelength; k is the Boltzmann constant; $T_{R}$ is the system noise temperature; and B is the equivalent bandwidth of the receiver.

Bistatic radar differs from monostatic radar in the following ways: (1) The contour lines of the detection range of bistatic radar are Cassini oval lines, while those of single-base radar are circles. (2) The contour lines of the bistatic range are elliptical and do not align with the detection contour lines, while those of monostatic radar are aligned. (3) Bistatic range cell width varies with target range and azimuth, while that of monostatic radar is fixed. (4) The resolution of bistatic targets is related to the bistatic angle. (5) The receiving and transmitting direction maps and propagation factors of bistatic radars may be different, while those of single-base radars are the same [10].

The following considers the line-of-sight constraint observation range. Taking into account the curvature of the Earth’s surface, the target seen by the bistatic radar must be in simultaneous line of sight of the transmitting and receiving stations. If the target height is $h_{t}$, the receiving antenna height is $h_{R}$, and the transmitting antenna height is $h_{T}$, using the 4/3 ellipsoid model of the Earth, the radii $r_{T}$ and $r_{R}$ of the two covering circles are approximately as follows:(3)$$r_{T}=4.12(\sqrt{h_{t}}+\sqrt{h_{T}})$$(4)$$r_{R}=4.12(\sqrt{h_{t}}+\sqrt{h_{R}})$$

In the formula, the units of $h_{T}$, $h_{R}$, and $h_{t}$ are meters, and the units of $r_{T},r_{R}$ are kilometers. In this page, $h_{T}$ and $h_{R}$ are set to 15 m and 20 m, $h_{t}$ is 10 m. The bistatic detection range under line-of-sight constraints is the common part of the two circular coverage areas, as shown in Figure 1. The area can be expressed as follows:(5)$$A_{c}=\frac{1}{2}\left[r_{R}^{2}\left(\varphi_{R}-\sin\varphi_{R}\right)+r_{T}^{2}\left(\varphi_{T}-\sin\varphi_{T}\right)\right]$$

The $\varphi_{R}$ and $\varphi_{T}$ in the formula are, respectively:(6)$$\varphi_{R}=2\cos^{-1}\left(\frac{r_{R}^{2}-r_{T}^{2}+L^{2}}{2r_{R}L}\right)$$(7)$$\varphi_{T}=2\cos^{-1}\left(\frac{r_{T}^{2}-r_{R}^{2}+L^{2}}{2r_{T}L}\right)$$

#### 2.1.2. Multistatic Radar Positioning Method

By combining data like signal time, phase, and angle, multistatic radar systems usually use the cooperation of several transmitting and receiving stations to position targets. Nowadays, the following positioning techniques are typically used in real-world applications:(1)Positioning methods based on time information, which measure time information such as the Time Difference of Arrival (TDOA) and the distance between two bases through multistatic radar systems. In TDOA, the time difference of signal arrival is measured by multiple receiving stations, and the intersection point of the hyperbola is formed to determine the target position. This method requires high-precision time synchronization and is suitable for distributed radar systems. In the bistatic distance and method, the total distance from the transmitting station to the target and then to the receiving station (bistatic distance) is utilized. Multiple bistatic distances form an elliptical trajectory, and the target is located through the intersection of the ellipses [11].(2)Angle of Arrival (AOA) and other angular information-based positioning techniques use the intersection of geometric triangles to estimate the position of each receiving station, which measures the incident angle of the target signal. High-precision direction-finding devices, including phased array antennas, are necessary for this technique [12].(3)The Doppler frequency shift-based positioning technique uses data from several receiving stations along with the difference in Doppler frequency shift brought on by the target’s movement to determine the target’s position and speed. This method is mostly applicable to the dynamic target tracking of radar transmission [13].(4)By integrating TDOA with AOA and combining temporal information with angular information for hybrid positioning, the robustness of positioning is enhanced and geometric dilution errors are reduced [14].

In this paper, the signal-to-noise ratio is calculated by applying the bistatic radar equation to determine whether the target is detectable. The target is located through geometric models using information such as distance sum, baseline distance and angle, and geometric formulas.

According to the bistatic radar equation, the signal-to-noise ratio of the received signal is the same as Equation (2).(8)$$SNR=\frac{P_{T}G_{T}G_{R}\lambda^{2}\sigma }{{\left(4\pi \right)}^{3}kTBL_{T}L_{R}L_{s}R_{T}^{2}R_{R}^{2}}$$

Here, $P_{T}$, $G_{T},G_{R}$, λ, k, T, B are as defined previously in Equation (2). $L_{T},L_{R}$ represent total system losses (transmitter + receiver), including propagation losses. σ is the bistatic RCS of the target, which depends on the bistatic angle β, frequency, and target aspect angle. $R_{T}$ and $R_{R}$ are the distances from the transmitter to the target and from the target to the receiver. $L_{s}$ is the synchronization loss factor to quantify the SNR degradation due to time/frequency synchronization errors. The core goal of this research is to explore the effectiveness of receiver placement optimization methods rather than to develop signal processing algorithms. Therefore, we adopt the classical radar equation as the first approximation in the system-level performance analysis. We assume that the critical parameters (such as carrier frequency and approximate bandwidth) of the opportunistic illumination source have been obtained through an advanced third-party Signal Intelligence (SIGINT) database or a prior reconnaissance phase.

The geometric relationship of the bistatic radar system is shown in Figure 2. When the baseline distance L, sum of distances $R_{R}+R_{T}$, and the receiving angle $\theta_{R}$ are known, by the cosine theorem, we can obtain the following:(9)$${\left(R_{T}+R_{R}\right)}^{2}=L^{2}+{\left(R_{R}-R_{T}\right)}^{2}+4R_{T}R_{R}\sin^{2}(\frac{\beta }{2})$$

After simplification, we obtain the following:(10)$$R_{R}=\frac{{\left(R_{R}+R_{T}\right)}^{2}-L^{2}}{2(R_{R}+R_{T}+L\sin\theta_{R})}$$(11)$$R_{T}=\sqrt{R_{R}^{2}+L^{2}-2R_{R}L\cos(\frac{\pi }{2}+\theta_{R})}$$

Then, the double base angle β of the transmitter—target—receiver is as follows:(12)$$\beta =\arccos(\frac{R_{R}^{2}+R_{T}^{2}-L^{2}}{2R_{R}R_{T}})$$

In addition, three factors should be considered during the positioning process.

(1)Time error

Scanning antenna refers to the mechanical scanning antenna of the opportunistic irradiation source (Tx). In this study, the passive receiver uses staring antenna beams to continuously monitor the entire airspace of interest and simultaneously receive signals from multiple opportunistic illumination sources in different directions. The scanning antenna characteristics mentioned in this paper, such as beam width and scanning rate, are only used to calculate the dwell time of the transmitted signal on the target and then determine the number of coherent accumulation pulses. The time error $\Delta R_{time}$ is caused by the antenna beam width $\theta_{beam}$ and the scanning rate:(13)$$\Delta R_{time}=\frac{\theta_{beam}\cdot R_{T}}{2\sqrt{3}}$$

Further combine geometric factors:(14)$$\Delta R_{time}=(\frac{1+e^{2}+2e\sin\theta_{R}}{2{\left(1+e\sin\theta_{R}\right)}^{2}})\cdot \frac{\theta_{beam}}{2}R_{T}$$

Here, $e=\frac{L}{R_{R}+R_{T}}$ represents the eccentricity.

(2)Baseline error:

In the ship positioning and navigation system, the positioning accuracy and dynamic update delay of AIS (Automatic Identification System) are two key parameters that affect the positioning performance of multistatic radars [15]. Ship positions are obtained via the AIS system using satellite navigation systems such as GPS. The greatest difference between the reported position and the actual physical position is referred to as its positioning accuracy. Real-time updates are not made to AIS data; the dynamic update delay is the amount of time that passes between the vessel’s real location change and the new position that the AIS system reports. The positioning accuracy of AIS and the dynamic update delay jointly cause the baseline error $\Delta R_{L}$:(15)$$\Delta R_{L}=\left|\frac{-(e^{2}+1)\sin\theta_{R}-2e}{2{\left(1+e\sin\theta_{R}\right)}^{2}}\right|\cdot (\Delta L_{AIS}+\Delta L_{delay})$$

(3)Azimuth error:

The azimuth error $\Delta R_{\theta }$ is caused by the direction-finding accuracy of ESMs (Electronic Support Measures). The direction-finding accuracy of ESM refers to the degree of accuracy of the system in measuring the Direction of Arrival (DOA) of the signal. It is usually expressed as an angular error (such as ±1.5°). This parameter directly affects the accuracy of target positioning in systems such as radar and communication countermeasures:(16)$$\Delta R_{\theta }=\left|\frac{L(1-e^{2})\cos\theta_{R}}{2{\left(1+e\sin\theta_{R}\right)}^{2}}\right|\cdot R_{T}\Delta \theta_{R}$$

The total error is calculated from the Root Sum Squares (RSS) of each error source:(17)$$\Delta R_{total}=\sqrt{{\Delta R_{time}}^{2}+{\Delta R_{L}}^{2}+{\Delta R_{\theta }}^{2}}$$

### 2.2. Non-Dominated Sorting Genetic Algorithm

The NSGA-II is a widely used multi-objective evolutionary algorithm known for its effectiveness and efficiency [16,17,18]. Its core operations—non-dominated sorting, crowding distance calculation, and elitism—enable it to find a diverse set of solutions approximating the Pareto-optimal front for problems with conflicting objectives.

#### 2.2.1. The Core Principle of Non-Dominated Sorting Genetic Algorithm

Solution $x^{*}$ is called the Pareto optimal solution if and only if(18)$$\exists x\in X:\{\begin{array}{c} f_{i}(x)\leq f_{i}\left(x^{*}\right)\forall i\in \{1,2,3\}\\ f_{j}(x)<f_{j}\left(x^{*}\right)\mathrm{at}\;\mathrm{least}\;\mathrm{one}\;j \end{array}$$

Figure 3 shows the basic structure of the Non-dominated Sorting Genetic Algorithm.

#### 2.2.2. Multistatic Radar Optimization

In the optimization of multistatic radar systems, the objective function model is established first. The detection coverage rate calculation is as follows:(19)$$f_{1}(x)=\sum_{k=1}^{K}A_{k}\cdot P_{d}(p_{k})$$

Here, A represents the detection coverage area. The detection probability is modeled using the Marcum Q-function for a non-coherent integrator detecting a Swerling I target model.(20)$$P_{d}=Q(\sqrt{2\cdot SNR\cdot N},\;\sqrt{-2lnP_{f_{a}}})$$

Among them, Q is the Marcum Q function, which is the core of statistical detection. SNR is the signal-to-noise ratio of Equation (8), and $P_{f_{a}}$ is the false alarm probability, N is the number of pulses integrated non-coherently. For the purposes of coverage calculation in this optimization, a target is considered detectable if $P_{d}$ exceeds a predefined threshold (e.g., 0.8) for a given $P_{f_{a}}$.

The Cramér-Rao Lower Bound (CRLB) provides a lower bound on the variance of any unbiased estimator of the target position. For a hybrid TDOA/AOA measurement system, the Fisher Information Matrix (FIM) F for estimating the target position $x={\left[x,y\right]}^{T}$ is given by the following:(21)$$F={\sum_{i=1}^{N}\frac{1}{\sigma_{\tau_{i}}^{2}}{\left(\frac{\partial_{\tau_{i}}}{\partial_{x}}\right)}^{T}\left(\frac{\partial_{\tau_{i}}}{\partial_{x}}\right)+\sum_{i=1}^{N}\frac{1}{\sigma_{\theta_{i}}^{2}}\left(\frac{\partial_{\theta_{i}}}{\partial_{x}}\right)}^{T}\left(\frac{\partial_{\theta_{i}}}{\partial_{x}}\right)$$

Among them, $\tau_{i}$ is the TDOA measurement for the ith transmitter-receiver pair; $\theta_{i}$ is the AOA measurement at the receiver for the signal from the ith transmitter; the gradients $\frac{\partial_{\tau_{i}}}{\partial_{x}}$ and $\frac{\partial_{\theta_{i}}}{\partial_{x}}$ are the Jacobians relating the measurements to the target position. ${\sigma_{\tau_{i}}}^{2},\;{\sigma_{\theta_{i}}}^{2}$ are the variances of the TDOA and AOA measurements, respectively. These values are critical and must be set based on realistic assumptions about the system’s capabilities. $\sigma_{\theta }$ is set to 1.5° according to the commonly used ESM system performance. $\sigma_{\tau }$ is related to the system bandwidth B and SNR, it is approximated by $\sigma_{\tau }\approx \frac{1}{B\sqrt{\mathrm{SNR}}}$. Based on the parameters in Table 1 and assuming typical SNR operating points, $\sigma_{\tau }$ takes a value of about 0.1 μs.

Then, the CRLB for the position estimate is as follows:(22)$$CRLB=\sqrt{trace(F^{-1})}$$

Secondly, carry out constraint processing:

Angle constraint:(23)$$|\theta_{i}-\theta_{j}|\geq \Delta_{\min}$$

Baseline constraint:(24)$$L_{\min}\leq L\leq L_{\max}$$

Adopt the penalty function method:(25)$$\phi (x)=f(x)+\rho \sum_{j}max{\left(0,g_{j}\;(x)\right)}^{k}$$
where f(x) is the original objective function, ρ is the penalty coefficient, $g_{j}(x)$ represents the j TH inequality constraint, including the angle and baseline constraints mentioned above, and k is the penalty index, usually set to k=2. In the application of this paper, the original probe coverage is used as the original objective function, and a new objective function is generated under the angle and baseline constraints, so as to optimize the probing performance.

According to the application scenario of multistatic radar system optimization, some improvements and innovations are made to the algorithm. The initialization strategy based on radar characteristics is adopted, and the initial population is generated by using Cassini’s oval line characteristics.(26)$$R_{t}R_{r}=b^{2}\Rightarrow p=(\frac{b^{2}cos\theta }{a(1+ecos\theta )},\;\frac{b^{2}sin\theta }{a(1+ecos\theta )})$$

## 3. Results

### 3.1. Simulation Process

The multistatic radar system used in this paper consists of one receiver and several transmitters, the detection target is a general cargo ship, and the scene representation is shown in Figure 4.

The SNR is determined using the bistatic radar equation and the MATLAB (R2023a) model of the marine scene. The SNR is then used to determine whether the target was detected. In order to replicate the random distribution features of merchant ships in the real scene, four transmitters were positioned at random within 25 km of the baseline distance using a random function, with the receiver serving as the origin point. The receiver deployment site was improved to enhance detection performance, while the receiver location was optimized using the multi-objective genetic algorithm. Monte Carlo simulation was carried out 50 times, and the detection performance and distribution law before and after optimization were compared to determine the optimal deployment strategy of the receiver. Under the premise of given bistatic incidence angle and receiving angle, the RCS value is extracted from the bistatic RCS table in reference [3] by the cubic spline interpolation method, which effectively improves the simulation efficiency. The interpolation table has been provided in the Appendix A. In this paper, the S-band radar is selected as the transmitter, and its parameters are shown in Table 1; the algorithm parameters are shown in Table 2. The four transmitters, while all operating in the S-band, are modeled with parameters that can have minor variations around the typical values listed in Table 1, reflecting real-world conditions.

### 3.2. Analysis of Simulation Results

After computing the average positioning error and detection coverage across 50 optimization simulation cycles, a bar chart was generated to facilitate comparison between the initial and optimized results. Figure 5 shows the Pareto front obtained from the optimization run, illustrating the trade-off between detection area and positioning error. In the figure, the 50 points with different colors represent the Pareto front in 50 simulations, and the five-pointed stars represent the Pareto optimal solution in each simulation. Figure 6a shows the average detection area before and after optimization, Figure 6b shows the average positioning error before and after optimization. According to the results, the optimized detection area increased by about 15.8% (404 km^2^), and the average positioning error decreased by about 8.9% (7 m).

In order to find the optimal location of the deployed receiver more intuitively, the position comparison between the receiver and the geometric center of the transmitter before and after five optimization simulations is randomly drawn, as shown in Figure 7. It is found that the simulation in which the receiver is closer to the geometric center of the transmitter takes up a large proportion after optimization, which suggests that the system’s detection performance is influenced by the distance between the receiver and the transmitter’s geometric center.

To explore the influence of this factor, the distance between the receiver and the geometric center is taken as the horizontal axis, and the number of occurrences of the optimized receiver within a certain distance is taken as the vertical axis for plotting, as shown in Figure 8. The red dash line marks the boundary at the 15 km threshold. After optimization, 72% of the receiver positions are located within 15 km of the geometric center. In addition, the trend of the detection performance as a function of the receiver position is plotted, as shown in Figure 9. When the optimized receiver is located within 15 km of the geometric center, the detection area of the system is between 2800 km and 3150 km, and the positioning error range is about 70 m. When the distance between the receiver and the geometric center exceeds 15 km, the detection performance of the system is all degraded, which is also proven by the fitting curve in Figure 9.

## 4. Discussion

In the current electromagnetic environment, non-cooperative passive radar has several benefits and possible applications. In this paper, a multi-objective genetic algorithm is used to optimize the receiver and transmitter layout of a multistatic passive radar system, which provides a new idea and method for the application of a non-cooperative passive multistatic radar system in reality.

Under the S-band radar parameters used in this paper, the comparison of multiple data and simulation results show that using the multi-objective genetic algorithm to optimize the receiver position can improve the detection range of the non-cooperative passive multistatic radar system by about 404 km^2^, reduce the positioning error by about 7 m; compared with the random placement baseline, they are 15.8% and 8.9% higher and improve the system detection performance significantly. According to the law of the optimized receiver position and the comparison of the simulation data, in the actual application scenario, the receiver is placed within 15 km of the geometric center of the geometry composed of four transmitters, which can significantly improve the detection range of the system and is more likely to achieve better detection performance.

This study demonstrates the viability of PMR networks as a powerful complement to traditional active monostatic systems. While a dedicated monostatic radar may achieve superior range and positioning accuracy for a single target under ideal conditions, the optimized PMR network offers compelling alternative advantages. These include significantly lower cost and energy consumption by leveraging existing illuminators, enhanced resilience through spatial diversity and the absence of a single point of failure, and inherent covertness due to its passive nature. Therefore, the value proposition of PMR lies not in outright outperforming active radar in every metric, but in providing a cost-effective, robust, and stealthy solution for persistent maritime domain awareness, where its performance, as shown, is within acceptable operational standards.

As shown in Table 3, our NSGA-II-based approach demonstrates competitive performance compared to other methods. While convex relaxation methods provide convergence guarantees, they often require problem simplification that may lead to suboptimal solutions in complex non-convex scenarios like ours. MOEA/2 and SPEA2 offer good diversity but at a higher computational cost. Our method achieves a favorable balance between coverage improvement and error reduction, which is particularly suited for passive multistatic radar systems with geometric constraints.

For future work, the inclusion of noise and clutter suppression links in the algorithm design is considered. In addition, based on the non-cooperative constraint, the merchant ships as transmitters are randomly distributed and move. Therefore, the detection of low RCS targets and the construction of dynamic transmitter network architecture will become the focus of research in the future work, so that the simulation environment is closer to the actual application scenario, so as to enhance the convincing of the simulation results. Another key direction for future research is to extend the optimization framework to include more pronounced transmitter heterogeneity, such as a mix of S-band, VHF, and satellite-based illuminators of opportunity, to fully exploit the potential of diverse opportunistic sensor networks.

## Figures and Tables

**Figure 1 sensors-25-05861-f001:**
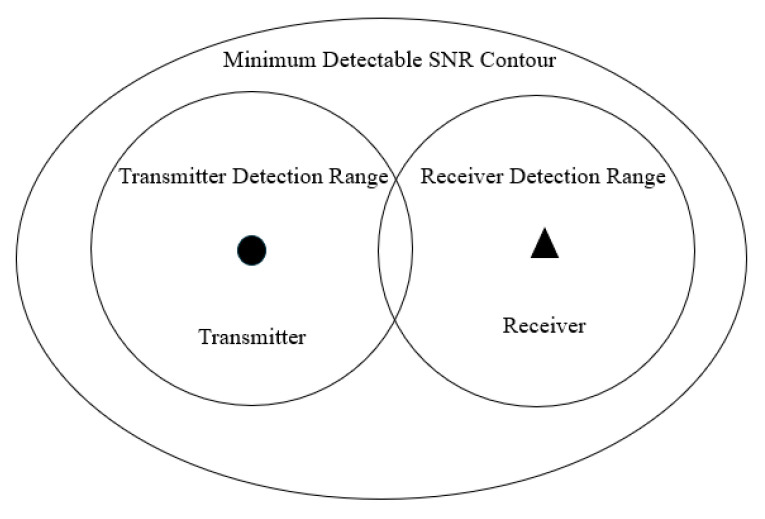
Bistatic Radar Coverage Area.

**Figure 2 sensors-25-05861-f002:**
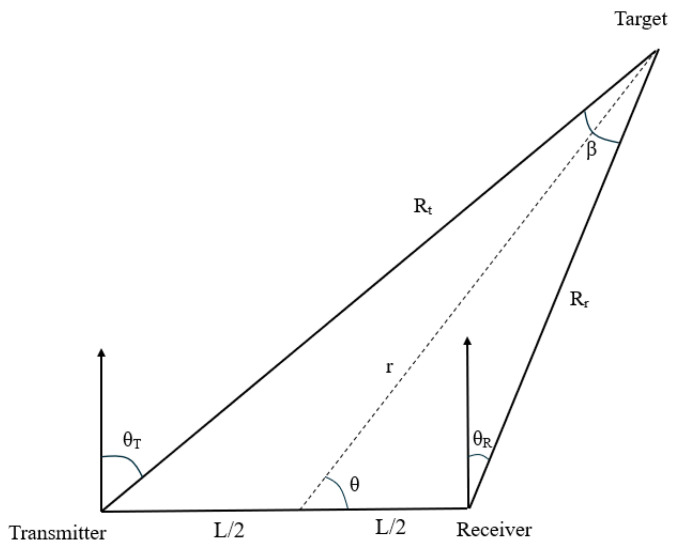
Multistatic radar polar coordinate geometry diagram.

**Figure 3 sensors-25-05861-f003:**
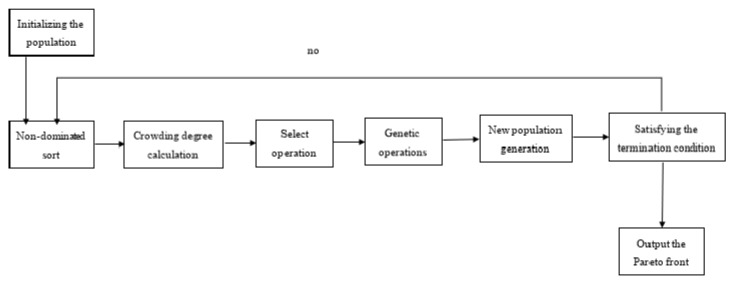
The basic structure of the Non-dominated Sorting Genetic Algorithm.

**Figure 4 sensors-25-05861-f004:**
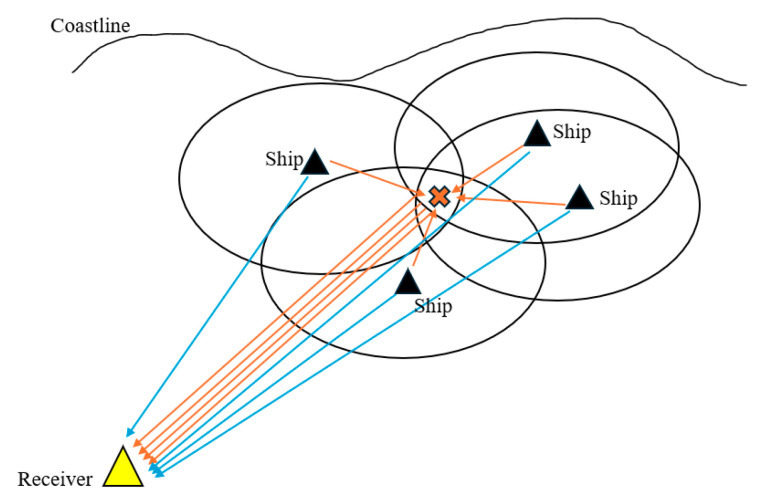
Multistatic Radar Scenario Diagram.

**Figure 5 sensors-25-05861-f005:**
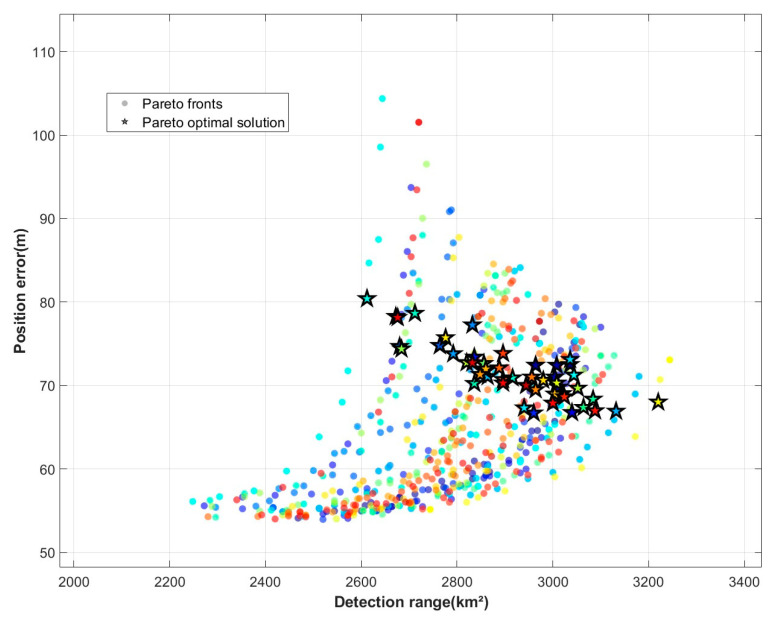
Pareto optimal solution and Pareto front.

**Figure 6 sensors-25-05861-f006:**
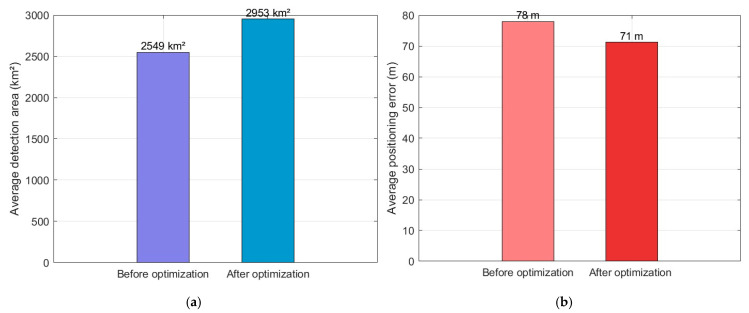
Comparison of detection performance before and after optimization. (**a**) Comparison of average coverage area before and after optimization. (**b**) Comparison of average positioning error before and after optimization.

**Figure 7 sensors-25-05861-f007:**
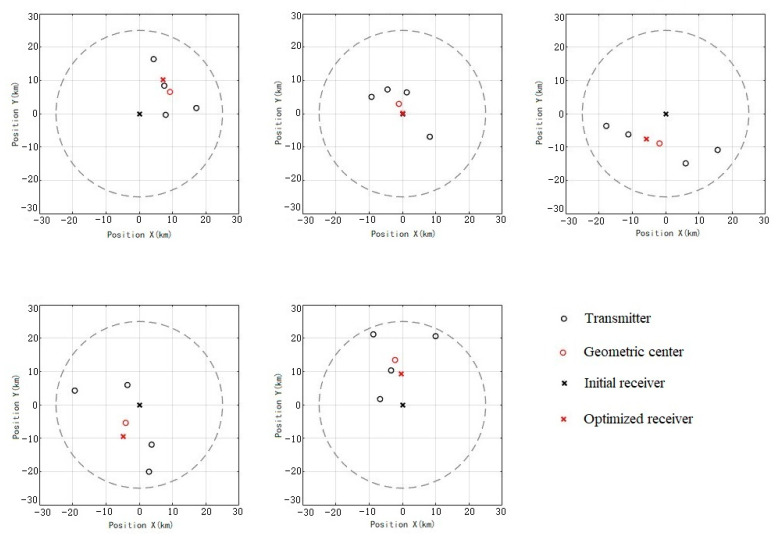
Comparison of receiver position before and after optimization.

**Figure 8 sensors-25-05861-f008:**
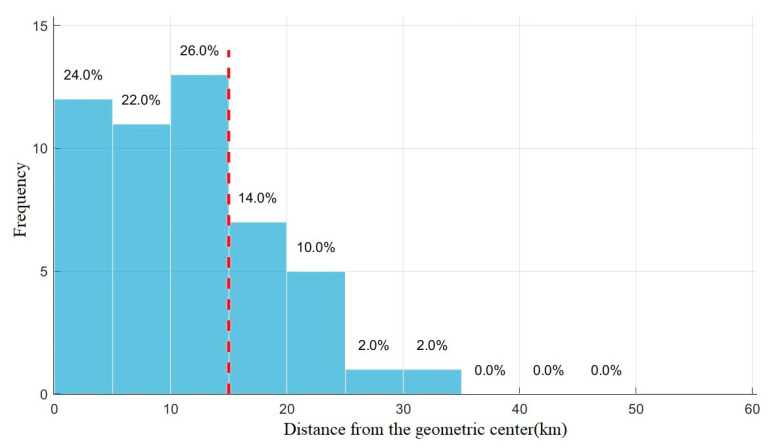
Frequency of occurrence of different distances between the receiver and the geometric center after optimization.

**Figure 9 sensors-25-05861-f009:**
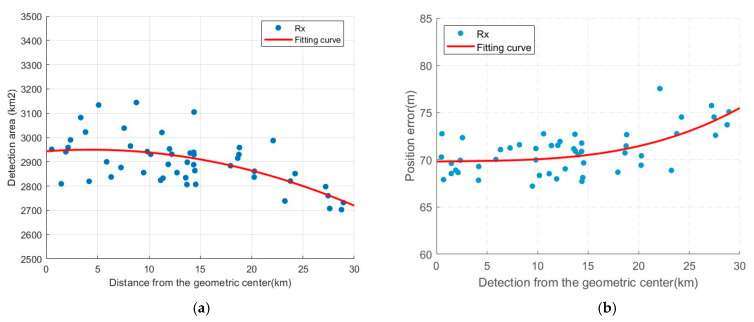
Detection performance as a function of receiver position: (**a**) Detection area as a function of receiver position; (**b**) Position error as a function of receiver position.

**Table 1 sensors-25-05861-t001:** Radar parameters.

Serial Number	Radar Parameters	Measurement Value
1	Width of beam	1.8°
2	Scan rate	30 rpm
3	Power output	30 kw
4	Frequency	3000 MHz
5	Gain of antenna	28 dB
6	Wavelength	0.0995 m
7	Bandwidth of noise	5 MHz
8	Pulse repetition frequency	1000 Hz

**Table 2 sensors-25-05861-t002:** Algorithm parameters.

Serial Number	Algorithm Parameters	Measurement Value
1	Population size	50
2	Generations	50
3	Crossover rates	0.8
4	Migration rates	0.2
5	Bounds	Geometric center ± 25 km
6	Encoding	Real encoding
7	Penalty coefficient	50

**Table 3 sensors-25-05861-t003:** Comparison to modern receiver-placement/geometry-design literature for multistatic localization.

Method	Optimization Objectives	Constraints	Gains
Convex Relaxation [19]	GDOP minimization	Linear/Convex	~15–20% GDOP reduction
MOEA/D [20]	Coverage, Accuracy	Geometric	~15–20% coverage improvement
SPEA2 [21]	Detection probability, Positioning error	Energy, Geometric	~10–16% error reduction
Gradient-based [22]	CRLB minimization	Differentiable constraints	~12–18% CRLB improvement
Proposed NSGA-II	Coverage area, Localization accuracy	Bistatic angle, Baseline distance	15.8% coverage gain, 8.9% error reduction

## Data Availability

The data in this study are not publicly available or shared due to the military sensitivity of them.

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
