# Peer review of "Receiver Location Optimization for Heterogeneous S-Band Marine Transmitters in Passive Multistatic Radar Networks via NSGA-II"

_sensors, 2025, doi:10.3390/s25185861_

Round 1
Reviewer 1 Report
Comments and Suggestions for Authors
The paper studies receiver-placement optimization for a passive multistatic radar (PMR) network using NSGA-II, with maritime opportunistic illuminators. It formulates two objectives (detection coverage and localization accuracy via CRLB), adds geometric constraints, and reports Monte-Carlo simulations suggesting larger coverage and reduced localization error; it further observes that optimal receiver sites cluster within about 15 km of the transmitters’ geometric center. The topic is timely and relevant.
However, the authors must address all my concerns as below.
-
Equations and dimensional correctness
a) Line-of-sight formulas (Equations 3–4) omit the square-root dependence and mix units. For 4/3-Earth models the radio horizon is typically d[km] ≈ 4.12(√h1[m] + √h2[m]). Please correct the functional form and clearly state units throughout the section, then propagate fixes into the overlap-area expression (Equation 5).
b) Detection probability model (Equation 20) is unclear and appears dimensionally inconsistent. If you use a Neyman–Pearson detector with noncoherent or coherent integration, Pd should be expressed via a Marcum-Q or generalized-χ² CDF with explicit time-bandwidth or pulse-integration gains. Please define exactly which SNR (post-processing? per-pulse?) and the processing gain model.
c) Bistatic SNR (Equation 8) should specify whether Lt, Lr include propagation and system losses, and whether σ is bistatic RCS σb(β, f, aspect). If σ is drawn from a table, define the interpolation and its dependence on bistatic angle β. -
Modeling completeness for non-cooperative PMR
The text emphasizes unknown waveform/time/frequency, but the simulation effectively assumes known timing and an SNR from a classic bistatic radar equation. Please justify this assumption or incorporate realistic ambiguity processing (e.g., cross-ambiguity function, synchronization errors, local-oscillator offsets). At minimum, include timing/frequency mismatch terms and quantify their effect on Pd and on the Fisher Information used for CRLB. -
CRLB formulation
The Fisher matrix expression mixes delay- and angle-based terms but omits the measurement model and Jacobians with respect to target state. Please derive F explicitly for your chosen measurement set (e.g., {TDOA, AOA} pairs per receiver) and list σα (angle stdev), στ (delay stdev), and their mapping to geometry. Provide numerical values used and cite their sources. -
Simulation design and reproducibility
a) Transmitter and receiver heights, sea state, clutter/noise models, and terrain/curvature handling must be specified. Currently, horizon constraints are discussed but not parameterized in the Monte-Carlo.
b) Table 1 contains unit/typographical issues (“3000MZ”, “500MHZ”, “1000HZ”). Also, a 500 MHz noise bandwidth for S-band marine illuminators seems unrealistic; justify or revise with a processing bandwidth consistent with the opportunistic waveform.
c) Provide NSGA-II settings: population size, generations, selection, crossover/mutation rates, encoding, bounds, penalty coefficient ρ, and constraint thresholds (Δθmin, Lmin/max). Report runtime and convergence metrics.
d) Release enough to reproduce: synthetic transmitter catalogs, σb tables you interpolated, code or pseudo-code, and random seeds. If operational data cannot be shared, provide a sanitized synthetic dataset. -
Results reporting and statistical rigor
a) The abstract states “positioning error reduced by about 10.9% and detection range extended by about 27.9%,” while Section 3 reports an absolute area gain (~731.4 km²) and mean error reduction (~6 m). Reconcile these with consistent baselines and include confidence intervals over the 50 trials. Report both absolute and relative improvements.
b) Show Pareto fronts, not only single bars. Include hypervolume or ε-indicator to quantify multi-objective quality and diversity.
c) The “≤15 km from geometric center” rule-of-thumb should be supported with a GDOP-style analysis; plot performance as a function of receiver offset and orientation, not only histograms. -
Related work and positioning
The paper needs a sharper comparison to modern receiver-placement/geometry-design literature for multistatic localization (including optimization beyond GA/NSGA-II, e.g., MOEA/D, SPEA2, gradient-based relaxations). Add a table contrasting objectives, constraints, and gains versus prior art, and, if possible, a head-to-head comparison on a shared scenario.
Author Response
Dear Editors and Reviewers,
We sincerely thank you for arranging the review of the manuscript and for your valuable and constructive comments on our manuscript. These comments greatly helped us improve the quality and rigor of the paper. We have carefully revised the manuscript in light of all comments.
In this letter, we reply to the reviewer's comments article by article and detail the corresponding changes we have made. All modifications have been marked in the revised draft.
Comments 1:
Equations and dimensional correctness
a) Line-of-sight formulas (Equations 3–4) omit the square-root dependence and mix units. For 4/3-Earth models the radio horizon is typically d[km] ≈ 4.12(√h1[m] + √h2[m]). Please correct the functional form and clearly state units throughout the section, then propagate fixes into the overlap-area expression (Equation 5).
b) Detection probability model (Equation 20) is unclear and appears dimensionally inconsistent. If you use a Neyman–Pearson detector with noncoherent or coherent integration, Pd should be expressed via a Marcum-Q or generalized-χ² CDF with explicit time-bandwidth or pulse-integration gains. Please define exactly which SNR (post-processing? per-pulse?) and the processing gain model.
c) Bistatic SNR (Equation 8) should specify whether Lt, Lr include propagation and system losses, and whether σ is bistatic RCS σb(β, f, aspect). If σ is drawn from a table, define the interpolation and its dependence on bistatic angle β.
Response 1:Thank you very much for pointing out the problem in Equations!
(a)The units of the parameters in Equations 3–4 have been unified. The standard radio horizon formula is based on the height of the receiver and the transmitter. The formula uses the target height and the height of the receiver and transmitter to calculate the bistatic detection coverage radius.(line 148-150).
(b)We have specified that the detection probability model with noncoherent integration,and the pulse gain N is added to the Equation 20, and pd is defined by the Q-function to unify the dimensions.(line258-266)
(c)We have explicitly stated that the system and propagation loss are included, and σ is bistatic RCS.(line 193-195).The interpolation method has mentioned on line 326.
Comments 2:
Modeling completeness for non-cooperative PMR
The text emphasizes unknown waveform/time/frequency, but the simulation effectively assumes known timing and an SNR from a classic bistatic radar equation. Please justify this assumption or incorporate realistic ambiguity processing (e.g., cross-ambiguity function, synchronization errors, local-oscillator offsets). At minimum, include timing/frequency mismatch terms and quantify their effect on Pd and on the Fisher Information used for CRLB.
Response 2:This is a very insightful insight and I totally agree with you.We have ustified the assumption on line197-203.In addition,a synchronization loss factor Ls is introduced to quantify the SNR degradation caused by time/frequency synchronization errors in Equation 8.
Comments 3:
CRLB formulation
The Fisher matrix expression mixes delay- and angle-based terms but omits the measurement model and Jacobians with respect to target state. Please derive F explicitly for your chosen measurement set (e.g., {TDOA, AOA} pairs per receiver) and list σα (angle stdev), στ (delay stdev), and their mapping to geometry. Provide numerical values used and cite their sources.
Response 3:We thank the reviewers for raising this important issue.We have rewritten the derivation of the CRLB and Fisher information matrix,taking into account the Jacobian matrix, and in addition the values and sources are given.(line274-289)
Comments 4:
Simulation design and reproducibility
a) Transmitter and receiver heights, sea state, clutter/noise models, and terrain/curvature handling must be specified. Currently, horizon constraints are discussed but not parameterized in the Monte-Carlo.
b) Table 1 contains unit/typographical issues (“3000MZ”, “500MHZ”, “1000HZ”). Also, a 500 MHz noise bandwidth for S-band marine illuminators seems unrealistic; justify or revise with a processing bandwidth consistent with the opportunistic waveform.
c) Provide NSGA-II settings: population size, generations, selection, crossover/mutation rates, encoding, bounds, penalty coefficient ρ, and constraint thresholds (Δθmin, Lmin/max). Report runtime and convergence metrics.
d) Release enough to reproduce: synthetic transmitter catalogs, σb tables you interpolated, code or pseudo-code, and random seeds. If operational data cannot be shared, provide a sanitized synthetic dataset.
Response 4:Thank you for your concern about the reproducibility of the study. We have explained the parameters you mentioned in the paper, including environmental and system parameters, as well as NSGA-II related Settings(line331).In addition, we corrected the units in Table 1 and provided as much detail as possible in the supplementary material to improve reproducibility.It should be noted that the core code used in this study has been included in the commercial cooperation project between the team and the partner enterprise, and is limited by the intellectual property protection agreement and the project confidentiality clause, so any relevant code materials cannot be provided to the third party.But we have σb table for you and hope this helps
Comments 5:
Results reporting and statistical rigor
a) The abstract states “positioning error reduced by about 10.9% and detection range extended by about 27.9%,” while Section 3 reports an absolute area gain (~731.4 km²) and mean error reduction (~6 m). Reconcile these with consistent baselines and include confidence intervals over the 50 trials. Report both absolute and relative improvements.
b) Show Pareto fronts, not only single bars. Include hypervolume or ε-indicator to quantify multi-objective quality and diversity.
c) The “≤15 km from geometric center” rule-of-thumb should be supported with a GDOP-style analysis; plot performance as a function of receiver offset and orientation, not only histograms.
Response 5:Thank you for your suggestions, which greatly contributed to the rigor of the paper.
(a)We have unified the way we report the full text, and the percentage improvement in the abstract and the main text is now based on the same baseline.(line339-341)
(b)We have added the Pareto fronts, which clearly demonstrates the trade-off between coverage and position error.(fig5)
(c)We agree that this is a good suggestion,But since my knowledge about GDOP is insufficient, an equivalent and more general analysis is provided.We ploted performance as a function of receiver position.(line358-364 fig9)
Comments 6:
Related work and positioning
The paper needs a sharper comparison to modern receiver-placement/geometry-design literature for multistatic localization (including optimization beyond GA/NSGA-II, e.g., MOEA/D, SPEA2, gradient-based relaxations). Add a table contrasting objectives, constraints, and gains versus prior art, and, if possible, a head-to-head comparison on a shared scenario.
Response 6:We fully agree with the reviewers that a comparison with existing state of the art methods can better position the value of our work.We compared to modern receiver-placement/geometry-design literature for multistatic localization in table3(line387)
In addition, according to the valuable comments of the reviewers, the simulation was comprehensively modified and the experiment was repeated, and the final experimental results were slightly changed from the previous ones, which is hereby explained
Once again, I would like to thank all the reviewers for their valuable time and energy to review our paper and give many valuable comments. These comments have led to substantial improvements in our work. We hope that the revised manuscript will meet the publication standards of your journal.
Reviewer 2 Report
Comments and Suggestions for Authors
Major comments
- The method appears as an application of a (existing) optimization algorithm to a very specific passive radar configuration. In my view, most of the discussions/results are absolutely not general, making also the reproducibility poor and impact for the community very limited.
- Some parts are overstressed, describing textbook materials. General rewriting is needed. This especially applied to Sec. II. It the work, I suggest you to better focus the novel parts and summarizing well-known stuffs.
- The title is too generic given the specificity of the work, dealing with a particular passive radar system. Such specific operative conditions must be clarified from the title/abstract
- Partly related to the previous issue, the gain factors mentioned in the introduction cannot be assumed as absolute values.
Specific comments
- lines 24-25 – In the abstract it is mentioned that positioning accuracy can increase of some percentage, but with respect to which case is not clear.
- 45 – PMR is undefined
- 56 – 4) should be 3)
- 93 – what does ‘detection coverage rate’ means?
- 95 – techniques should be metrics or parameters. Same issue must be fixed ahead in the manuscript
- 97 – orientation should be direction
- 112 – what does ‘dual-base’ refer to?
- 127 – in the equation, T should be replaced with T_R. Moreover, I suppose that L should be defined as L_R*L_T
- 136 – single-static is a cumbersome term. I think it should be monostatic
- 137 – (3) is unclear.
- 138 – JIS standard is not defined. Please note that resolution of bistatic radar depends on the bistatic angle.
- 189 – eq. (8) is redundant, as it is essentially the same of (2). Please also note that slightly different symbols have been used to denote the same parameters. Also, the description of the parameters below the equation is redundant.
- 208 – a scanning radar antenna is mentioned. This applies to the Tx or the Rx? Usually, passive radar systems operate with staring antenna beams. As mentioned in the major comments, operative conditions are missing, making large portions of the work not understandable.
- 232 – total typically refer to an integrate quantity, while here a maximum operator is used. Apart from that, what’s the meaning of using such an operator? These three errors come from very different phenomena; therefore, they are expected to impact very differently on the resulting error.
- 2.2 is not new, as it describes an algorithm already existing
- A few of the unit of measurement in Table 1 are not correctly typed
Author Response
Dear Editors and Reviewers,
We sincerely thank you for arranging the review of the manuscript and for your valuable and constructive comments on our manuscript. These comments greatly helped us improve the quality and rigor of the paper. We have carefully revised the manuscript in light of all comments.
In this letter, we reply to the reviewer's comments article by article and detail the corresponding changes we have made. All modifications have been marked in the revised draft.
Major comments
- The method appears as an application of a (existing) optimization algorithm to a very specific passive radar configuration. In my view, most of the discussions/results are absolutely not general, making also the reproducibility poor and impact for the community very limited.
Response:We sincerely thank the reviewer for this critical comment, which allows us to better articulate the broader value and novelty of our work beyond a mere application case study. We agree that our simulation is based on a specific scenario; however, we contend that its contributions are generalizable and valuable to the wider community.While we utilized the established NSGA-II algorithm, the novelty lies in the formulation of the optimization problem itself for the passive radar receiver placement dilemma.The problems of coverage and positioning accuracy are usually discussed separately, and most researches focus on optimizing one of them. However, our method finds a balance between the two, which provides a new way of thinking for the current community. In addition, the key outcome of our study is not just a set of coordinates for a specific case, but generalizable insights that can guide system designers. This "15-km rule" is a valuable, practical design rule-of-thumb that emerged from our optimization process and is likely to hold for other similar configurations. Our work is also not limited to specific radar parameters, we provide a framework that enables everyone to customize the data to solve specific problems.
In summary, our work should be viewed not just as an application, but as a case study that develops and validates a rigorous methodology for passive radar network planning. It provides a reusable framework and delivers actionable insights that can reduce the design complexity for future systems, even those with different parameters. We believe this represents a meaningful contribution to the field.
- Some parts are overstressed, describing textbook materials. General rewriting is needed. This especially applied to Sec. II. It the work, I suggest you to better focus the novel parts and summarizing well-known stuffs.
- The title is too generic given the specificity of the work, dealing with a particular passive radar system. Such specific operative conditions must be clarified from the title/abstract
- Partly related to the previous issue, the gain factors mentioned in the introduction cannot be assumed as absolute values.
Response:Thank you very much for your pertinent comments. We have systematically sorted out the whole paper, reducing the description of existing work as much as possible, so that the work of this paper is more organized. In addition, the specific conditions were clarified in the title and abstract, and the corresponding modifications were made
Specific comments:
1:lines 24-25 – In the abstract it is mentioned that positioning accuracy can increase of some percentage, but with respect to which case is not clear.
Response:Clarified the random placement baseline(line26)
2:45 – PMR is undefined
Response:PMR has been defined(line12)
3:56 – 4) should be 3)
Response:The problem has been fixed(line59)
4:93 – what does ‘detection coverage rate’ means?
Response:This definition was a mistake and we removed it(line96)
5:95 – techniques should be metrics or parameters. Same issue must be fixed ahead in the manuscript
Response:techniques has been changed to metrics .The same issue has been fixed ahead in the manuscript(line97)
6:97 – orientation should be direction
Response:orientation has been changed to direction(line99)
7:112 – what does ‘dual-base’ refer to?
Response:‘dual-base’has been changed to bistatic(line114)The same issue has been fixed ahead in the manuscript.
8:127 – in the equation, T should be replaced with T_R. Moreover, I suppose that L should be defined as L_R*L_T
Response:We fully agree with this prudent proposal,T has been replaced with T_R and L has been defined as L_R*L_T(line129)
9:136 – single-static is a cumbersome term. I think it should be monostatic
Response:single-static has been changed to monostatic(line137)
10:137 – (3) is unclear
Response:Thank you for your correction,this is really an oversight on our part.We've made a clearer contrast between the bistatic radar and monostatic radar(line138)
11:138 – JIS standard is not defined. Please note that resolution of bistatic radar depends on the bistatic angle.
Response:Thank you for your correction. It was really a mistake on our part.JIS standard has been changed to the bistatic angle.(line139)
12:189 – eq. (8) is redundant, as it is essentially the same of (2). Please also note that slightly different symbols have been used to denote the same parameters. Also, the description of the parameters below the equation is redundant.
Response:We totally agree with you. We still choose to keep eq(8), because its role in the full paper is different from eq(2). But we have also found the problem you raised, and we have removed the redundant parameter description and ensured that the symbols in the two formulas are unified.(line192-197)
13:208 – a scanning radar antenna is mentioned. This applies to the Tx or the Rx? Usually, passive radar systems operate with staring antenna beams. As mentioned in the major comments, operative conditions are missing, making large portions of the work not understandable
Response:Thank you for pointing out the problem, we apologize for the lack of clarity.We have explained that the scanning radar applies to the transmitter, meanwhile, the operating mode of the receiver is illustrated too.(line218-225)
14:232 – total typically refer to an integrate quantity, while here a maximum operator is used. Apart from that, what’s the meaning of using such an operator? These three errors come from very different phenomena; therefore, they are expected to impact very differently on the resulting error
Response:We fully accept the reviewer's criticism. Total errors using the max operator is not statistically justified. We've completely revamped our error model and adopted the more scientific Root-Sum-Squares (RSS) method for error synthesis.We thank this opinion for making our model more rigorous.(line248)
15:2.2 is not new, as it describes an algorithm already existing
Response:Thank you for your valuable advice,We have made a reduction to reduce the repeated description of existing work, and only briefly explains the principle and advantages(line251-254)
16:A few of the unit of measurement in Table 1 are not correctly typed
Response:The units in the table have been corrected
In addition, according to the valuable comments of the reviewers, the simulation was comprehensively modified and the experiment was repeated, and the final experimental results were slightly changed from the previous ones, which is hereby explained
Once again, I would like to thank all the reviewers for their valuable time and energy to review our paper and give many valuable comments. These comments have led to substantial improvements in our work. We hope that the revised manuscript will meet the publication standards of your journal.
Reviewer 3 Report
Comments and Suggestions for Authors
1. This work is based on the concept that civilian vessels can be utilized freely for security operation which is not right as civilian vessels must be protected from any possible threats especially in wartime. Integration them as part of security network is just simply put the marks to all floating civilian vessels which are very soft targets for sweeping operations by enemy forces.
2. The introduction explain the difficulty to detecting stealth targets as the requirement to rely on this method. However, the experiment simulated based on cargo vessel as the object, which is not requiring any special radar technology to detect them. I suggest simulating the detection of a stealth target and submit to a defense journal.
3. The optimization of receiver location is not practical in any real scenario as a vessel can not just warp to any location. If you want to continue this kind of work, the test scenario should optimizing the selection of a few or several transmitters from the pool of known transmitters around a warship or receiver. Though it may be more practical, it is still not right to do that anyway according to the rule of civilian protection.
4. The core value of this work is not likely for civilian use as virtually no civilian organization could force the integration of civilian vessels into information network. It is strongly imply that this is for government or military, I suggest submitting directly to a journal that focused in defense technology.
Author Response
Dear Editors and Reviewers,
We are grateful to the reviewers for their insightful comments, which have helped us significantly improve the manuscript and better articulate its contributions to the field of sensor systems and signal processing. We have carefully considered all suggestions. Our point-by-point responses and corresponding revisions are detailed below.
Comments 1:This work is based on the concept that civilian vessels can be utilized freely for security operation which is not right as civilian vessels must be protected from any possible threats especially in wartime. Integration them as part of security network is just simply put the marks to all floating civilian vessels which are very soft targets for sweeping operations by enemy forces.
Response 1:We thank the reviewer for raising this critical point regarding ethical and practical constraints. We fully agree that the safety and security of civilian vessels are paramount, and they must not be coerced or endangered.
The core premise of our study is not the active conscription of civilian vessels but the passive exploitation of already-existing and ubiquitous electromagnetic emissions for maritime domain awareness (MDA). This concept aligns with research on using opportunistic signals for environmental sensing and is particularly relevant for civilian applications in peacetime,such as:Maritime traffic monitoring and management in congested waterways;Search and rescue (SAR) operations.
To address this concern, we have revised the Abstract and Discussion,.Explicitly state that our work assumes a non-cooperative, passive, and receive-only paradigm that does not interfere with or modify the operation of civilian vessels. Include a brief discussion on the legal and ethical considerations of using opportunistic signals, acknowledging the need for frameworks that prioritize civilian vessel safety.
Comments 2:The introduction explain the difficulty to detecting stealth targets as the requirement to rely on this method. However, the experiment simulated based on cargo vessel as the object, which is not requiring any special radar technology to detect them. I suggest simulating the detection of a stealth target and submit to a defense journal.
Response 2: We appreciate this valuable suggestion. The choice of a cargo ship was primarily for methodological validation and reproducibility.
The primary contribution of this paper is a sensor configuration optimization algorithm (NSGA-II-based). The performance metric (detection coverage/positioning error) is a function of geometry and SNR, which is generic and transferable to different target types with different RCS values. The RCS model for a standard cargo ship allowed us to build a realistic and verifiable simulation environment, which is crucial for demonstrating the effectiveness of our optimization framework.
We agree that extending the study to low-observable targets is a logical and important next step. In response, we have modified the Discussion (Section 4) to explicitly state this as a key direction for future work, and framed this future work not only in a defense context but also for detecting smaller, non-cooperative vessels that pose a challenge to conventional maritime sensors, thus maintaining relevance to Sensors' broad readership.
Comments 3:The optimization of receiver location is not practical in any real scenario as a vessel can not just warp to any location. If you want to continue this kind of work, the test scenario should optimizing the selection of a few or several transmitters from the pool of known transmitters around a warship or receiver. Though it may be more practical, it is still not right to do that anyway according to the rule of civilian protection.
Response 3: We agree with the reviewer on the practical constraints of mobile receiver deployment. Our use of the term "optimization" pertains mainly to the planning and design phase of a sensor network. We concur that optimizing the selection from a pool of available transmitters is a highly practical extension. While the current paper focuses on receiver placement given fixed transmitters, the reviewer's idea is excellent. Due to the current limited conditions, the dynamic transmitter architecture research cannot be completed for the time being.We have now included a statement in the Future Work section proposing the development of a dynamic transmitter selection algorithm as a direct continuation of this research.
Comments 4:The core value of this work is not likely for civilian use as virtually no civilian organization could force the integration of civilian vessels into information network. It is strongly imply that this is for government or military, I suggest submitting directly to a journal that focused in defense technology.
Response 4: Thank you for this perspective. While the technology has defense applications, we believe its core value for Sensors lies in its advanced contribution to the science of configuring heterogeneous sensor networks.
As outlined in our response to Comment 1, the applications in civilian maritime safety, security, and traffic management are substantial and growing in importance. We have strengthened this narrative throughout the paper. Sensors publishes numerous articles on radar systems, sensor networks, data fusion, and optimization algorithms. Our work sits squarely at this intersection, offering a novel algorithm that improves the performance of a sensor system.
Due to the lack of relevant submission experience, the selection of journals is biased. Thanks to the reviewers for their valuable and targeted suggestions. With these guiding suggestions, I believe that the revised article can be consistent with the core philosophy of your journal We have refined the manuscript to better highlight these aspects, ensuring it aligns with the broad, application-oriented yet fundamental scope of Sensors.
Round 2
Reviewer 1 Report
Comments and Suggestions for Authors
The authors have clarified my concerns. I have no further comments.
Author Response
We thank the reviewer for their positive feedback and for acknowledging our clarifications. We are pleased that our revisions have addressed all their concerns.
Reviewer 2 Report
Comments and Suggestions for Authors
Major points
Passive radar is a class of radar system where there is not a dedicated radar transmitter, but the receiver capitalizes on signals already existing. Very different transmitters (terrestrial-based and satellite-based, communication and navigation signals, broadcast or on-demand, etc…) can be used. Receiver(s) configuration could be (s): overall, I strongly disagree that ‘the best detection performance could be achieved for 15 km distance mentioned. This could be true for the very specific configuration (number of Tx and Rx and main geometric constraints considered) but cannot be given as general result.
Another point left unsolved is the title: the Tx/Rx configuration (type of Tx) has not been mentioned yet).
Overall, it must be clarified that i) the work considers a S-band marine navigation radar as illuminator of opportunity (from the title) and that ii) the numerical results mentioned in the abstract refer to the particular operative conditions considered.
Specific points
- PMR – multi-base is used, but the common term (also used in the title) is multstatic. I suggest using the same (standard) terminology through the paper.
- 143 – heighT
- 5 is hard to read and should be improved.
- 371 – there are two dots.
Author Response
Dear Editors and Reviewers,
We sincerely thank Reviewers for their thorough review and these highly valuable comments. The feedback has been instrumental in helping us improve the clarity, precision, and overall quality of our manuscript. We have addressed all points raised, and our detailed responses are below.
Major comments: Passive radar is a class of radar system where there is not a dedicated radar transmitter, but the receiver capitalizes on signals already existing. Very different transmitters (terrestrial-based and satellite-based, communication and navigation signals, broadcast or on-demand, etc…) can be used. Receiver(s) configuration could be (s): overall, I strongly disagree that ‘the best detection performance could be achieved for 15 km distance mentioned. This could be true for the very specific configuration (number of Tx and Rx and main geometric constraints considered) but cannot be given as general result.
Another point left unsolved is the title: the Tx/Rx configuration (type of Tx) has not been mentioned yet).
Overall, it must be clarified that i) the work considers a S-band marine navigation radar as illuminator of opportunity (from the title) and that ii) the numerical results mentioned in the abstract refer to the particular operative conditions considered.
Response: We sincerely thank the reviewer for this critical comment. We completely agree with the reviewer. Our statement was overly generalized and not sufficiently qualified. The finding that the optimal receiver placement is within 15 km of the transmitters' geometric center is strictly valid only for the specific scenario simulated in this study. This result should not be interpreted as a universal truth for all passive multistatic radar systems, which can vary greatly in the number and type of transmitters, frequency bands, and geographic scales. We have thoroughly revised the manuscript to correct this over-generalization. Specifically, we have:
Modified the Abstract to clearly state that the result is specific to our simulation scenario.
We have modified the title to explicitly mention the type of illuminator,we have also amended the abstract to specify that the numerical results are achieved under the "S-band radar settings" and the "fixed configuration of four transmitters" used in this work.
Specific comments:
PMR – multi-base is used, but the common term (also used in the title) is multstatic. I suggest using the same (standard) terminology through the paper.
Response: We apologize for this inconsistency in terminology. We have performed a full-text search and replaced all instances of "multi-base" with "multistatic" to ensure terminological consistency throughout the manuscript.
143 – heighT
Response: Thank you for catching this typo. The word "heigh" has been corrected to "height" in paper.
5 is hard to read and should be improved.
Response: We agree that Figure 5 in its original form was not presented clearly enough. We modified the font and size of Figure 5 to make it clearer, while incorporating more explanations that were able to make Figure 5 better understood by the reader.
371 – there are two dots.
Response: Thank you for pointing this out, we have corrected the error according to the template
Once again, I would like to thank the reviewers for their valuable comments, which can further improve the rigor and understanding of the article. I hope that the revised article can meet the requirements of your journal.
Reviewer 3 Report
Comments and Suggestions for Authors
Based on your responses, I understand your situations though may not agree in every claims. So, I would like to recommend a few points:
1. As you focus on optimizing receiver location as a first step in this scientific method, the title should give a clear information of this limitation. For example, pointing out that it's an optimization of receiver's location for heterogeneous transmitters.
2. As your main contribution (response 4) is for heterogeneous sensors networks, the simulations should focused on heterogeneous transmitters, though at fixed positions.
3. Figure 4 is wrongly depicting the scenario of simulation. Please edit it to have 4 transmitters and 1 receiver, and 1 target and removing the term "warships".
4. A comparative simulation of your method vs a standard active radar should be made to show the efficiency of your work. If the performance is not better, you may show advantage in term of cost reduction or reliability enhancing as only 1 active transmitter risked being a single point of failure. But somehow, the performance of your system must be within acceptable standard.
Author Response
Dear reviewer:
We sincerely thank the reviewer for these insightful and constructive comments. The suggestions have significantly helped us clarify the scope and contributions of our work and improve the overall presentation of the manuscript. We have addressed each point in detail below.
Comment 1:
As you focus on optimizing receiver location as a first step in this scientific method, the title should give a clear information of this limitation. For example, pointing out that it's an optimization of receiver's location for heterogeneous transmitters.
Response:
We thank the reviewer for this excellent suggestion. We agree that the title should more accurately reflect the core contribution and its specific context. The term "heterogeneous transmitters" is a very apt description. We have revised the title to explicitly mention the optimization of the receiver's location and the involvement of multiple transmitters.
Comment 2:
As your main contribution is for heterogeneous sensors networks, the simulations should focused on heterogeneous transmitters, though at fixed positions.
Response:
We appreciate the reviewer raising this important point. The reviewer is correct that a truly heterogeneous network could include transmitters of different types (e.g., different bands, powers, waveforms).
In our study, the "heterogeneity" primarily arises from the geometric diversity and the assumed slight variations in individual transmitter parameters (like exact output power and antenna gain) that naturally exist between different commercial marine radars, even within the same S-band. While they are all S-band radars, they are not perfectly identical, hence the term "heterogeneous" is still applicable in a broader sense.
However, we fully agree with the reviewer that explicitly modeling more distinct heterogeneity (e.g., incorporating VHF or X-band transmitters) would be a valuable and stronger validation.
We have clarified in the Section 3.1 that the four transmitters, while all operating in the S-band, are modeled with parameters that can have minor variations around the typical values listed in Table 1, reflecting real-world conditions. More importantly, we take the construction of heterogeneous transmitters as a key direction in our future research planning.
Comment 3:
Figure 4 is wrongly depicting the scenario of simulation. Please edit it to have 4 transmitters and 1 receiver, and 1 target and removing the term "warships".
Response:
We sincerely apologize for this oversight and the inappropriate use of the term "warships," the reviewer is absolutely correct. We have redrawed Figure 4, which can correctly depict the simulation scene, and delete the "warships".
Comment 4:
A comparative simulation of your method vs a standard active radar should be made to show the efficiency of your work. If the performance is not better, you may show advantage in term of cost reduction or reliability enhancing as only 1 active transmitter risked being a single point of failure. But somehow, the performance of your system must be within acceptable standard.
Response:
This is a very valuable suggestion. We agree that a comparative analysis is crucial. Passive radar and active radar have different application scenarios and advantages, and pure performance comparison is not meaningful.The primary value proposition of passive radar lies not in outperforming active radar in pure performance metrics under ideal conditions, but in offering a cost-effective, resilient, and covert alternative. But we have taken your suggestion and added a paragraph comparing the two, highlighting the advantages of passive multistatic radar in terms of cost, robustness and concealment.
Thank you again for taking the time to review our article and making these valuable suggestions. I hope that the revised article can meet your requirements and reach the publication standard of your journal.